# Insights into the Movement and Diffusion Accumulation Characteristics of a Catastrophic Rock Avalanche Debris—A Case Study

Yifei Gong [1,2,3,4,5], Xiansen Xing [6], Yanan Li [4], Chun Zhu [2,3,6], Yanlin Li [4], Jianhua Yan [6], Huilin Le [6] and Xiaoshuang Li [1,7,*]

1   School of Urban Construction, Changzhou University, Changzhou 213164, China; gongyifei@emails.bjut.edu.cn
2   Key Laboratory of Geohazard, Fuzhou 610059, China; zhu.chun@hhu.edu.cn
3   Key Laboratory of Geohazard Prevention of Hilly Mountains, Ministry of Natural Resources of China, Fuzhou 100037, China
4   Key Laboratory of Urban Security and Disaster Engineering of Ministry of Education, Beijing University of Technology, Beijing 100124, China
5   Faculty of Land Resource Engineering, Kunming University of Science and Technology, Kunming 650500, China
6   School of Earth Sciences and Engineering, Hohai University, Nanjing 210098, China; xingxs@hhu.edu.cn (X.X.); yanjh@hhu.edu.cn (J.Y.); lehuilin@hhu.edu.cn (H.L.)
7   Key Laboratory of Rock Mechanics and Geohazards of Zhejiang Province, Shaoxing 312000, China; liyanan@emails.bjut.edu.cn (Y.L.); liyanlin955@emails.bjut.edu.cn (Y.L.)
*   Correspondence: xsli2011@cczu.edu.cn

**Abstract:** In this study, the 1991 rock avalanche, in Touzhai, Zhaotong, Yunnan, China, was considered the study object. The investigation of the landslide accumulation body revealed that the Touzhai rock avalanche accumulation body has the characteristics of wide gradation and poor sorting. A combination of field investigations, indoor and outdoor experiments, and numerical simulations were used to invert the occurrence and spreading range of rock avalanche-debris flow hazards. To invert and analyze its dynamics and the crushing process, a three-dimensional discrete element modeling was performed on the real terrain data. Simulation results showed that the movement time of the numerically simulated Touzhai rock avalanche was approximately 200 s. After 50 s of movement, the peak velocity reached 32 m/s, and the velocity gradually decayed after the sliding mass rubbed violently against the valley floor and collided with the mountain. Due to the meandering nature of the gully, the sliding mass makes its way down the gully and constantly collides with the mountain, making particles appear to climb, with some particles being blocked by the valley. After 150 s of movement, the average velocity rate decreased substantially, and the landslide-avalanche debris reached the mouth of the trench. After 200 s of movement, the average sliding velocity tends to 0 m/s, where the avalanche debris tends to stop and accumulate. When the rock avalanche movement reaches the mouth of the gully, the avalanche debris spreads to the sides as it is no longer bounded by the hills on either side of the narrow gully, eventually forming a 'trumpet-shaped' accumulation, and the granular flow simulation matched the findings of the landslide site accumulation.

**Keywords:** rock avalanche; numerical simulation; accumulation body; diffusion mechanism; dynamical fragmentation

## 1. Introduction

The Elm landslide-avalanche debris in Switzerland in 1881 was one of the world's earliest recorded rock avalanche events [1]. The flow transported 2232 m in 40–45 s at an average speed of 180 km/h, killing 115 people. In the 1963 Vaiont landslide in Italy, a gravity deformation of a deep rocky slope with a volume of approximately $2.75 \times 10^8$ m³





slid rapidly into the reservoir beneath at a speed of 60 km/h in 1 min, causing a flood that killed 1925 people downstream [2]. In 1987, the Val Pola avalanche debris ($4 \times 10^7$ m$^3$) in Valtellina, Italy, traveled down the ditch at over 200 km/h, destroying three villages along the way and killing 27 people [3].

In China, the 1983 high-speed landslide ($4 \times 10^7$ m$^3$) in Springhill, Gansu, crossed the Baxie River at a speed of 5–10 m/s and washed approximately 10 m of the opposite bank slope [4]. The entire transport path was approximately 0.8–1.0 km in length and lasted for 2 min, causing 220 deaths and 27 injuries, as well as an earthquake with a magnitude of approximately 1.4, which was recorded by the Lanzhou Seismological Observatory. In 2000, a landslide-avalanche debris occurred in Zamunong Gully, Yigong Township, Tibet [5]. After falling from an altitude of 5000 m, the voluminous slope ($3 \times 10^7$ m$^3$) formed high-speed avalanche debris by entraining debris material along the way, which transported 8–10 km in just 3 min, with an average accumulation thickness of 60 m [6]. In 2009, a landslide-avalanche debris event occurred at Jiwei Mountain, Chongqing, China, which resulted in the rapid destruction and disintegration of a rock mass with a volume of approximately $5 \times 10^6$ m$^3$, forming an accumulation area of approximately 2.2 km in length along the route, and resulting in 10 fatalities and 64 missing persons [7–9]. Landslide-avalanche debris are catastrophic, lethal, and destructive, is a major hazard to people's lives and property, and has been occurring continuously for nearly 100 years. The causes of its occurrence and the process of its accumulation have received much scholarly attention.

A distinctive feature of landslide-avalanche debris is that they spread over distances much greater than those inferred by any simple friction model. Several transport mechanisms have been proposed and explained for the long-range, high-speed characteristics of avalanche debris moving in low-gradient gully beds [10–13]. Shreve and Goguel [2,14,15] proposed 'air-cushion lubrication theory' and 'local vaporization theory' that could be summarized as the 'gas lubrication theory' in the landslide-avalanche debris migration and diffusion processes, which emphasizes the decisive role of gas in the rapid migration of debris. These high-speed, long-range dispersal mechanisms are important for specific landslide-avalanche debris events, but no universally accepted, uniform explanation exists.

As the visual object of field research, mound accumulation characteristics play a decisive role in the study of avalanche debris dispersion mechanisms. Landslide-avalanche debris accumulations have a wide range of grain size distribution, with fine particles of micron size and large masses of up to several meters in size [16–18]. The wide size distribution is a common feature of landslide-avalanche debris accumulations worldwide. In the investigation and analysis of the accumulation body, some scholars have reported that the accumulation body section has a 'reverse sequence' structural feature [19,20]. Cruden and Hungr [21] found that a sorting or phase separation phenomenon occurred in the section of Frank landslide accumulation in Canada, in which large particles remained above and fine particles moved below, which cannot be true. Vallance and Savage [22] explained this phenomenon by suggesting that inter-particle friction promotes particle sorting, with fine particles falling more readily into the spaces between coarse particles. Pudasaini [23] on the other hand, explained the particle sorting behavior based on a two-phase flow model, suggesting that the flow velocity of the pile varies with depth, which results in coarse particles vibrating in the upper layer and transporting them to the leading edge of the avalanche debris. Usually, the sample weight of a single sieving point is less than 20 kg, and the sample covers a small range of particle sizes; therefore, the results are not representative enough. This, to some extent, has limited the understanding of landslide-avalanche debris accumulation characteristics, affecting the judgment of avalanche debris dispersal mechanisms.

Particle flow experiment is one of the main research methods for high-speed and long-distance landslides. By establishing a simplified engineering geological model, it can be realized. Wang et al. [24] analyzed the remote effect of momentum transfer by tracking measurements of tracer particles on the surface of a granular flow based on a granular flow inclined trough experiment under three-dimensional topographic conditions and from a

quantitative point of view. Ge et al. [25] used PIV analysis to characterize the velocity field evolution of granular flow in a sloping channel. The existence of collision and momentum transfer between particles during the granular flow motion is revealed, and the momentum transfer mechanism during the high-speed remote landslide motion is explored.

Rapidly developing computer simulation techniques after 2010 has been a major boost towards the inversion of avalanche debris movement and dispersion processes. In order to simulate the mechanical behavior of discontinuous media such as rocks and soil particles, Cundall and Strack [26] proposed the discrete element method (DEM). Many scholars have achieved many results in avalanche debris using discrete elements. Tang [27] employed particle flow code (PFC$^{3D}$) to numerically simulate the whole process of the Jiwei Mountain landslide movement in Chongqing, China. The effects of different friction coefficients and bond strengths of particles on the farthest slip distance and accumulation of debris particles were mainly studied. Ge et al. [25,28] established a three-dimensional discrete element numerical model under the control of rock structure to investigate the motion evolution process and disaster-causing range of a high-speed, long run-out landslide, considering the example of the Jiwei Mountain landslide in Wulong, Chongqing, China. Liu et al. [29] proposed a three-dimensional discrete element model and a simulation method to simulate the whole process of initiation, movement, and accumulation of large landslides in Xinmo Village, Mao County, Sichuan Province. The reliability of the method was verified by comparing it with actual landslides, providing an efficient computational method for numerical simulation of high-speed, long run-out landslides and assessment of their disaster extent.

In the above research, it can be found that the difficulty of the research is the complexity of rock avalanche-debris flow and its huge volume is often accompanied by huge energy, and the occurrence of landslides is sudden [30,31]. The existing monitoring technology is often unable to obtain effective monitoring of the occurrence and movement process of rock avalanche-debris flow. At the same time, due to the limitation of the 'size effect', the conventional model test cannot make an in-depth exploration of its motion mechanism. Discrete element simulation has significant advantages in inverting the transport work and accumulation of rock avalanche-debris flow. However, a large number of previous studies are mostly two-dimensional models, which do not consider the energy dissipation effect of collision of three-dimensional terrain on particles, thus affecting the diffusion range of debris flow [32,33].

In this study, the Touzhai rock avalanche (TZRA) in Yunnan, China was taken as the research object. Through field investigation, remote sensing satellite image analysis and simulation comparison of discrete element method (particle flow code PFC$^{3D}$), the high speed, high energy, long distance and dispersion range were studied. In the numerical analysis, characteristics of the on-site accumulation body are used to verify the validity of the numerical model. The study focuses on the kinetic, fragmentation and influence of terrain on the whole process of avalanche debris movement characteristics of the TZRA, hoping to provide useful implications for the study of inversion of landslide avalanche debris impact range.

## 2. Study Area

### 2.1. Geological Background

The TZRA occurred in the Touzhai gully on the left bank of the Panhe River, Yunnan province, China approximately 30 km and 320 km from Zhaotong and Kunming cities, respectively, geographically located between 27°32′52″–27°34′15″N and 103°51′09″–103°52′50″E (Figure 1a) [34]. The Touzhai Gully is approximately 4.0 km long, with an average width of approximately 130 m at the bottom, a watershed area of 3.2 km$^2$, altitudes ranging from 1820 to 2940 m (Table 1), and a vertical difference of 1120 m (Figure 1b). The gully has a "V"-shaped cross-section, with an average variation in slope of 40.65°–52.00° on both sides [35]. The basin is structurally located in the northwest wing of the Panhe Fault and the Touzhai-Xindianzi Syncline, and the geological structure in the area is dominated by folds.

The Panhe Fault presents a northeast trend, with a width of the fault fracture zone of 50 m, and is composed of compressive cataclastic rock mass, showing compressive characteristics. Structural planes were developed in the Touzhai River Basin, and the structural planes controlling the development of landforms were mainly tensile and compressive-torsional structural planes. The headward erosion of the stream made the rock mass near the surface smaller, causing the unloading and rebounding of the compressive structural plane. The increase in the structural plane fissure also created conditions for precipitation and seepage, promoting weathering of the rock mass.

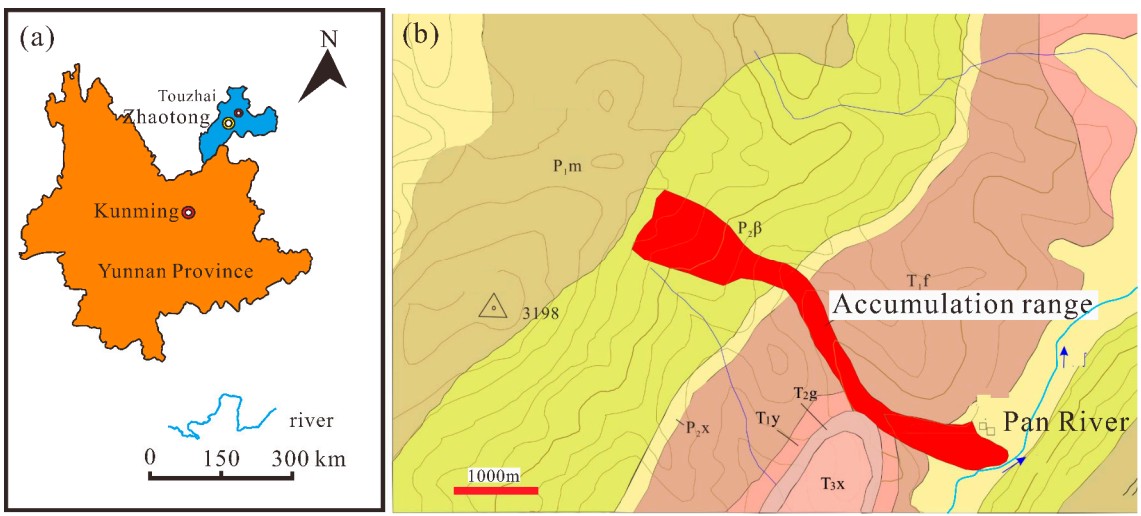

**Figure 1.** (**a**) Location of the study area; (**b**) Engineering geology of the study area.

**Table 1.** Main geomorphic geometric parameters of Touzhai River Basin [36].

| Basin Area (km²) | Highest Elevation (m) | Minimum Elevation (m) | Slope on Both Sides of the Valley (°) | Main Ditch Length (km) | Main Gully Channel Slope (°) |
|---|---|---|---|---|---|
| 3.2 | 2940 | 1820 | 40.65–52.00 | 4.0 | 12.4 |

Regarding stratigraphic lithology, the Permian and Triassic strata were mainly exposed in Touzhai. The following strata were exposed from the mouth of Touzhai to the source area of the rock avalanche: Lower Triassic Feixianguan Group; Lower Triassic Yongningzhen Group; Upper Permian Xuanwei Group and Permian Emeishan Basalt Group, with stratigraphic production of $170° \angle 45°$ in strike.

### 2.2. Climate

The study area was located in the warm zone and had a subtropical highland continental monsoon climate. Year-long automatic weather gauge monitoring data near the slip source area showed that the average temperatures in the hottest month (July) and coldest month are 16.5 °C and −1 °C, respectively, with an annual average temperature of 8.2 °C. Rainfall in the study area is mainly concentrated between May and September each year, with abundant rainfall, while rainfall is sparse between October and April. Based on the annual rainfall monitoring data, the rainfall in the source area is approximately 1.8 times the rainfall in the Touzhai area as monitored by the Zhaotong Meteorological Bureau, and rainfall varies both spatially and temporally [36].

### 2.3. Landslide Description

The site mapping (scale 1:1000) estimated that the volume of the source rock mass of the TZRA was approximately $9.0 \times 10^6$ m³, having a large slope instability. The unstable source rock started at the shear outlet at an elevation of 2300 m, collided with the gully scarp, and transformed into high-speed avalanche debris that transported 3.4 km in

3 min and finally stopped on the left bank of the Pan River. The average velocity of debris transport was approximately 68 km/h. The elevation of the back edge of the source area was 2580 m, and the elevation of the front edge of the accumulation was 1820 m. The apparent coefficient of friction was only 0.22 (H/L) (Figure 2), contributing to four major characteristics: large size, high speed, long distance, and low apparent coefficient of friction. The investigation demonstrated that the first half of the slide, after the main slide destabilized, quickly rushed out of the shear outlet and slid, violently hitting the slope of the left bank and climbing along the side, followed by the second half of the slide over both sides of the hill after the convergence with the first half of the slide, forming avalanche debris, and turning SE15° direction to spread. The debris were transported at high speeds and filled the valley floor and again rose up the slope of the right bank, eventually diving down the gully to the left bank of the Pan River.

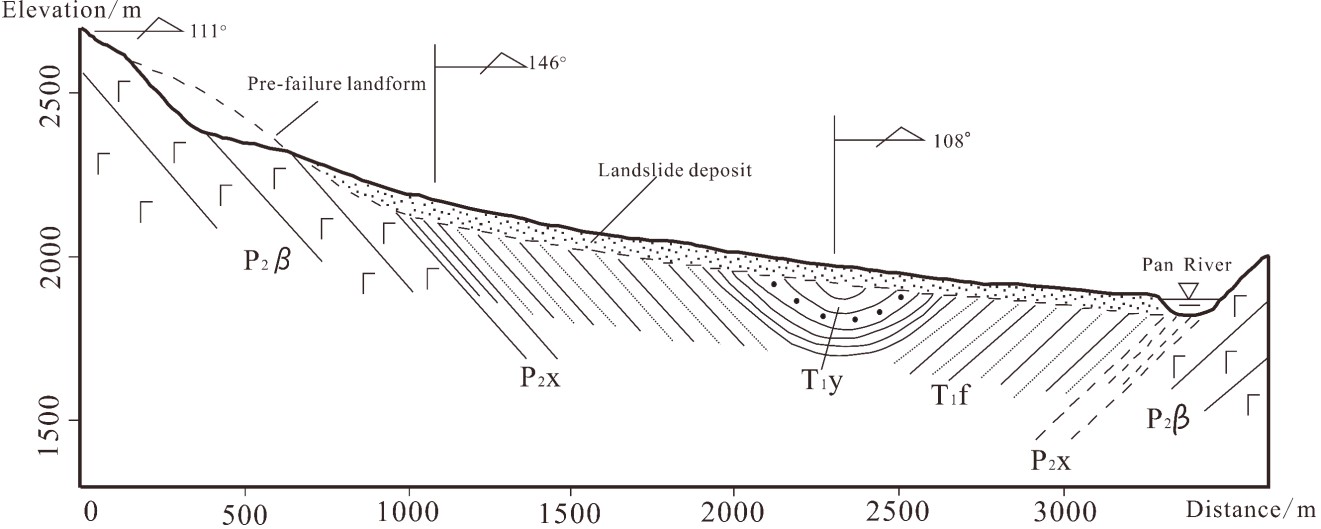

**Figure 2.** Longitudinal profile of the rock avalanche-avalanche debris sliding bed in Touzhai [36].

### 2.4. Deposit Features

Since the occurrence of the TZRA, the geomorphology has undergone great changes. The post-disaster ditch is filled with yellow debris soil and scattered boulders. As the topsoil of the mound is eroded to different degrees by surface water, two distinct landscapes emerge in the gully. After the surface layer of the landslide accumulation has been drained and eroded by groundwater, only angular basalt blocks with 5–20 cm diameter and good "sorting" remain in place. The valley terrain played a controlling role in determining the distribution of landslide deposits. By comparing the elevations at the same locations before and after the slip, we observed that the average thickness is approximately 10 m, the average width is 130 m, the width at the narrowest point is 60 m, and the widest point at the front edge is 230 m. The landslide deposits are mainly distributed in a band in the gully, which has no surface water flow in the dry season and surface runoff in the rainy season. During site research, the area once affected by the landslide avalanche debris can still be clearly seen. The scraping and shoveling effect of the landslide-avalanche debris movement on both sides of the mountain can be clearly seen through the different vegetation covers on the mountain. Only low herbaceous vegetation grows in the area affected by the avalanche debris, in contrast to the vegetation outside the impact area. Two boundaries, the main landslide-avalanche debris accumulation range and the ripple range boundary (shown in Figure 3a), can be roughly depicted by means of a fixed-point line on Google Earth, and the ripple range of avalanche debris can be observed far beyond the main accumulation range from the range curves of the two boundaries.

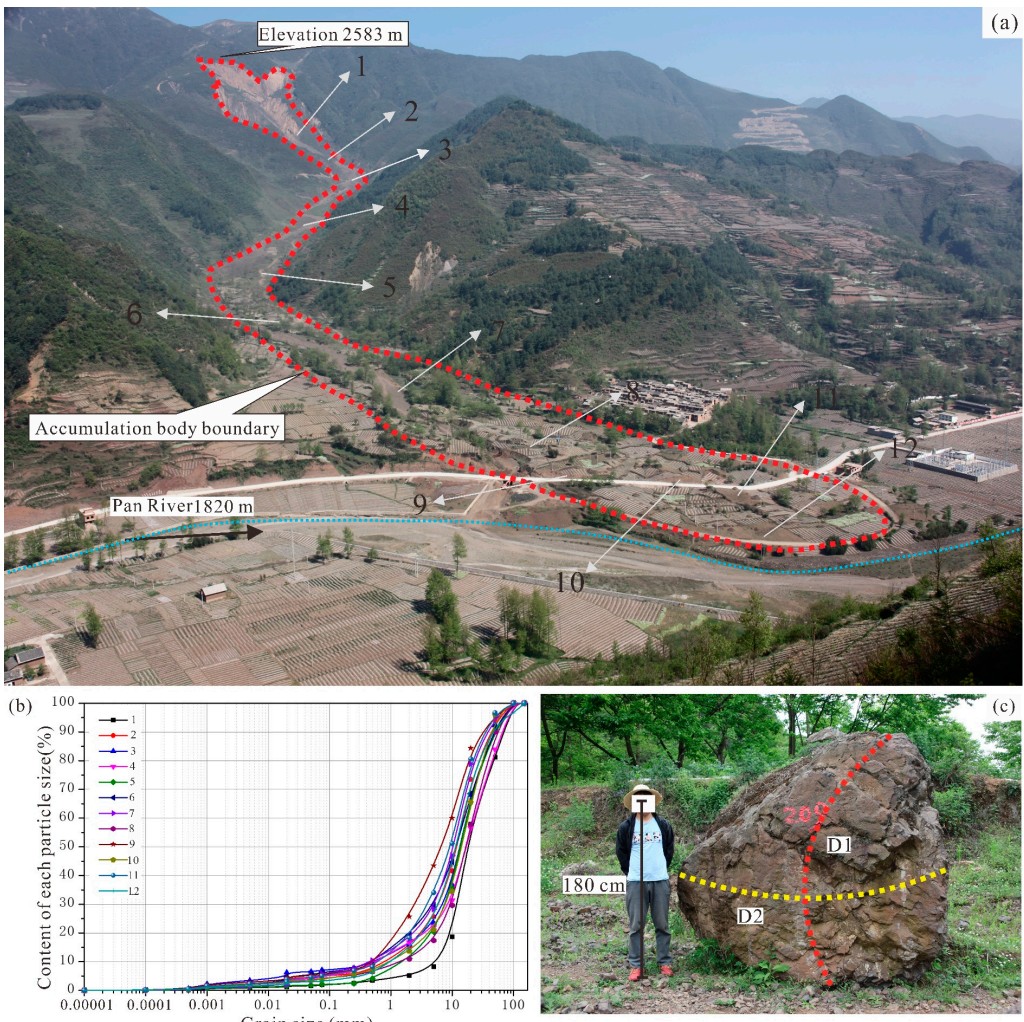

**Figure 3.** (**a**) Touzhai rock avalanche accumulation and influence ranges. (**b**) Grain size accumulation curve of the sampling points. (**c**) Equivalent particle size of block [35].

To facilitate the study of accumulation in Touzhai Gully, first, the gradation survey was conducted separately by particle size, and 12 points were distributed from upstream to downstream. The large span and high density of sampling made the analysis results representative. The gradation curves from the sieving points showed that the cumulative gradation curves were relatively similar, with the highest percentage of particles in the range of 20–50 mm reaching 23.79–39.29% (Figure 3b). Second, the accumulation of grain size above the meter level was investigated by using a combination of measurement, positioning, and geo-radar to investigate the boulder grain size and their accumulation location from the shear outlet to the location of the accumulation fan (Figure 3c).

The accumulation body of the TZRA is composed of 0.5~8 m boulders and gravel-clay mixtures below 20 cm, accounting for 10% and 90%, respectively. The boulders are distributed throughout the valley, and there are boulders with an equivalent particle size of approximately 3.5 m in the accumulation fan. A total of 992 boulders with equivalent particle size greater than 0.5 m were investigated on site, with an outcrop area of 2554 m$^2$, which is approximately 0.22% of the total debris accumulation area. GPS location (The longitude and latitude coordinates of the boulder are marked on Google Maps) of the boulders in the ditch was performed to obtain the distribution of boulders in the valley (Figure 4).

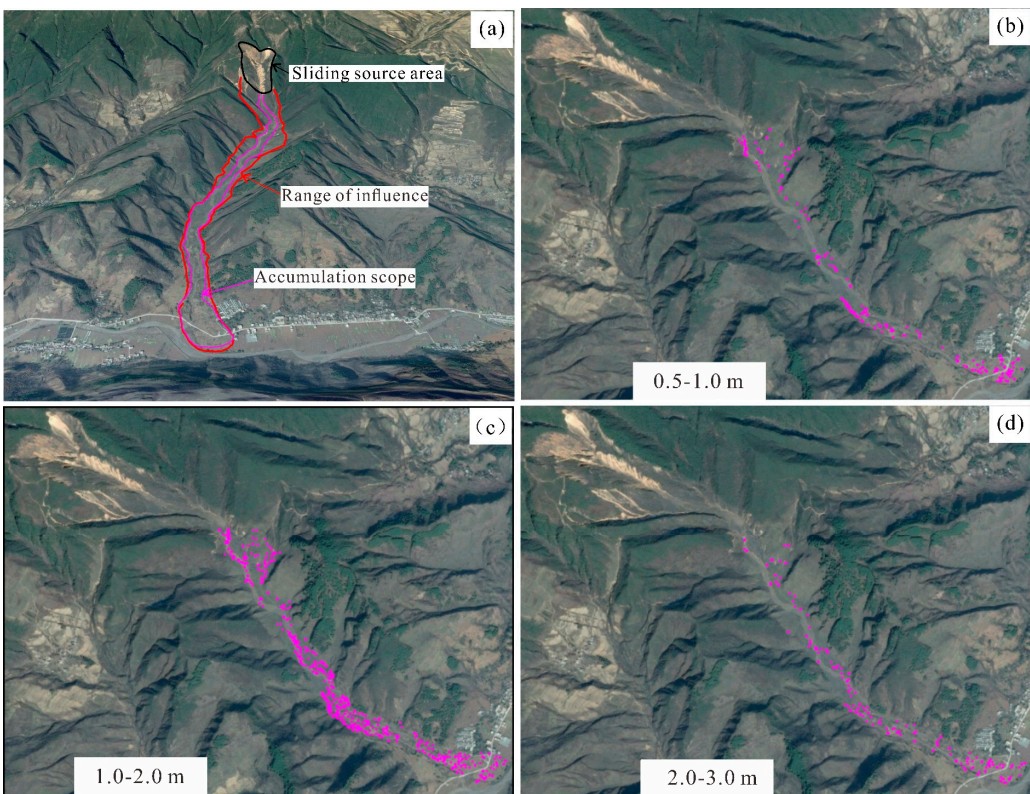

**Figure 4.** (**a**) Touzhai rock avalanche accumulation and influence ranges. (**b–d**) Distribution of each equivalent particle size of boulders [35].

## 3. Methodology

### 3.1. Discrete Element Modelling

To comprehensively reflect the whole process from the beginning of motion to the stopping of the accumulation of the TZRA. The motion time, average velocity, energy dissipation process, and characteristics of the final stopping motion of the TZRA were inverted using discrete element numerical software, PFC$^{3D}$ v.7. The scope of the model covers all stages of the entire TZRA movement. After determining the wave and influence range of the TZRA, an effective area was selected to establish a three-dimensional geological model of the TZRA avalanche debris (Figure 5a). The size of the completed numerical model of the TZRA was 3980 m long in the north-south direction and 1550 m wide in the east-west direction, and the maximum height difference of the model was approximately 800 m, which is consistent with the field topography. The three-dimensional model of the entire TZRA was composed of triangular walls (slip surface) with particle movement boundaries; each triangular surface was isosceles, with a total of 158,702 triangular surfaces and the volume of the landslide body in the source area was approximately $9 \times 10^6$ m$^3$. The slip surface can be modeled using wall groups, and the slip body uses approximately 12,000 balls with three groupings of radii 0.5–1.0 m, 1.0–2.0 m, and 2–3.0 m, which are uniformly filled in the slip source area. The linear bond contact model was used to simulate the rock mass between the spherical elements, and the linear contact model was used between the wall and sphere. In the linear bond model, the force and moment borne by the bond between particles are recorded as cementation behavior. When the applied stress exceeds its bond strength, the linear bond breaks the residual friction of the particles [37,38]. In this study, the TZRA was monitored for five main aspects: particle sliding mass velocity, particle movement time, slide energy, average displacement of the sliding mass, and particle accumulation range. To have a complete understanding of the TZRA movement characteristics on a spatial scale, the arrangement of monitoring points should consider two areas, the front edge of the landslide and the back edge of the landslide, respectively,

involving the surface particles and the bottom particles of the landslide mass (as shown in Figure 5b).

**Figure 5.** Model of the source rock mass in the Touzhai rock avalanche. (**a**) Three dimensional topographic map (**b**) Generated particles and filled areas.

### 3.2. Parameter Setting

The parameters in PFC used microscopic parameters between the particles because the macroscopic parameters in the actual geotechnical body hardly correspond to the microscopic parameters. The particles cannot be directly used to realize the macroscopic material properties, but there is a certain connection between the two. The common parameter calibration methods are the trial-and-error method and triaxial experiments to obtain stress-strain curves and compare them with macroscopic parameters (Table 2). However, the trial-and-error method involves many attempts, and the calibration parameters may cause relatively large errors. Therefore, the use of PFC to establish triaxial experiments for microscopic parameter calibration has become an effective tool for studying catastrophic landslides [39,40]. In the linear bond model, the effective bond modulus is related to Young's modulus, the particle stiffness ratio is related to Poisson's ratio, and the linear bond strength is related to the uniaxial compression stress; therefore, the servo mechanism in PFC$^{3D}$ was used to set up the uniaxial compression experiment without a lateral limit to obtain the stress-strain curve (Figure 6).

**Table 2.** Parameters of the sliding rock mass.

| Rock Mass | Density $\rho$ (g/cm$^3$) | UCS (MPa) | Young's Modulus (GPa) | Poisson's Ratio $\mu$ |
|---|---|---|---|---|
| Basalt | 2100 | 200 | 11 | 0.14 |

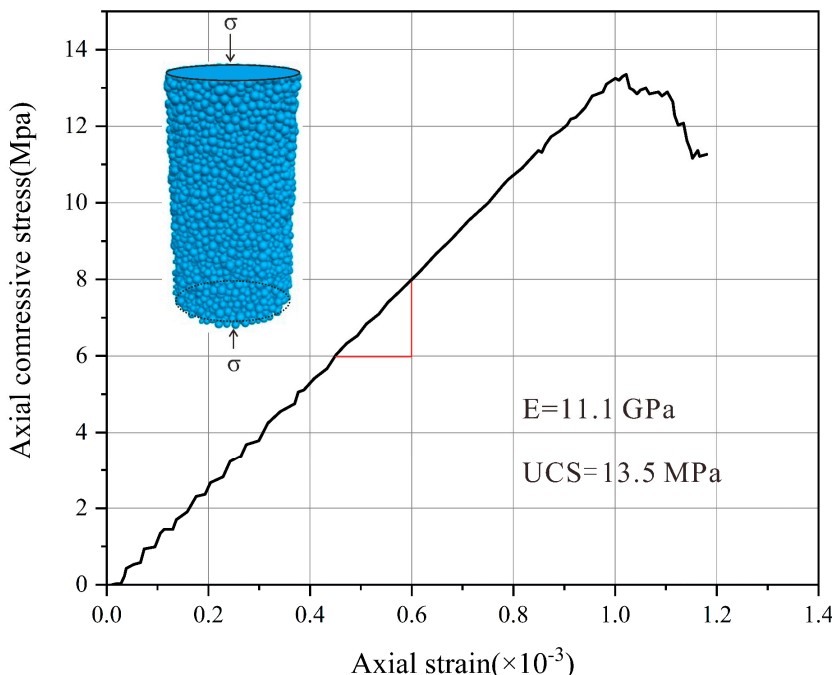

**Figure 6.** Uniaxial compression test.

The micro-parameters for the PFC model (Table 3) were derived from a numerical test of uniaxial compression (Figure 6), in which the unconfined compressive strength of rock mass UCSm (Uniaxial compression strength (MPa)), was fitted with that of Touzhai rock mass. UCSm of Touzhai rock mass was estimated to be 11.1 MPa, by an empirical relationship [41,42], where UCSr is the unconfined compressive strength of intact rock that was 200 MPa based on laboratory tests and S is an empirical parameter related to discontinuities in the rock mass that was estimated to be 0.004 according to Hoek et al. [43]. Based on the trial-and-error process, the best-fit macroscopic properties were obtained, and their relevant microscopic parameters are listed in Table 3.

**Table 3.** Numerical microscopic parameters of the numerical simulations.

| Parameter | Micro Parameter Type | Values |
|---|---|---|
| $R_{max}/R_{min}$ | Particle radius | 2 |
| $N$ | Number of particles | 87,000 |
| $\rho$ | Particle density (kg/m$^3$) | 2600 |
| $E_c$ | Ball-ball contact modulus (GPa) | 1.49 |
| $K$ | Normal-to-shear stiffness ratio (kn/ks) | 1.0 |
| $E'_c$ | Bond effective modulus (GPa) | 4.8 |
| $K'$ | Bond normal-to-shear Stiffness ratio | 1.2 |
| $\sigma_c$ | Contact-bond normal strength (MPa) | $5 \times 10^6$ |
| $\tau_c$ | Contact-bond shear strength (MPa) | $2 \times 10^6$ |
| $\mu_1$ | Friction coefficient (ball friction coefficient) | 0.3 |
| $\mu_2$ | Friction coefficient (wall friction coefficient) | 1.0 |

The particles in PFC are rigid substances, which cannot be used to simulate particle crushing and wearing, and the inelastic energy dissipation generated by the collision between particles is difficult to simulate directly. Therefore, damping was introduced to accelerate the convergence of the numerical solution and realize energy dissipation. Furthermore, the value of the viscous damping coefficients was obtained by back-analysis of the merical-experimental data [44]. Numerical simulations suggested that reasonable values of the viscous damping coefficients for the model were νn = νs = 0.05 (where νn and νs are the normal and shear damping constants, respectively).

## 4. Simulation Results

### 4.1. Simulation of Velocity Variations

After the initiation and accelerated motion of the TZRA, the rock mass rapidly crumbled and broke apart. The sliding body has experienced three collisions. After cutting out from the locking section, the sliding body first collided with the northwest slope of the watershed and the northwest slope at an azimuth angle of 111° and then collided with the left bank mountain at an azimuth angle of 90°. After that, the avalanche debris collided with the right bank slope and finally migrated to the gully along the azimuth angle of 108°. The simulation results showed that the TZRA movement time was approximately 200 s (Figure 7). The numerical simulation time was slightly longer than the actual observed duration of the TZRA event, which was 180 s. The main reason for this overestimation is that the discrete element software was set to dissipate the energy of each particle, and the velocity of motion tended towards zero, ending the calculation with more stringent conditions.

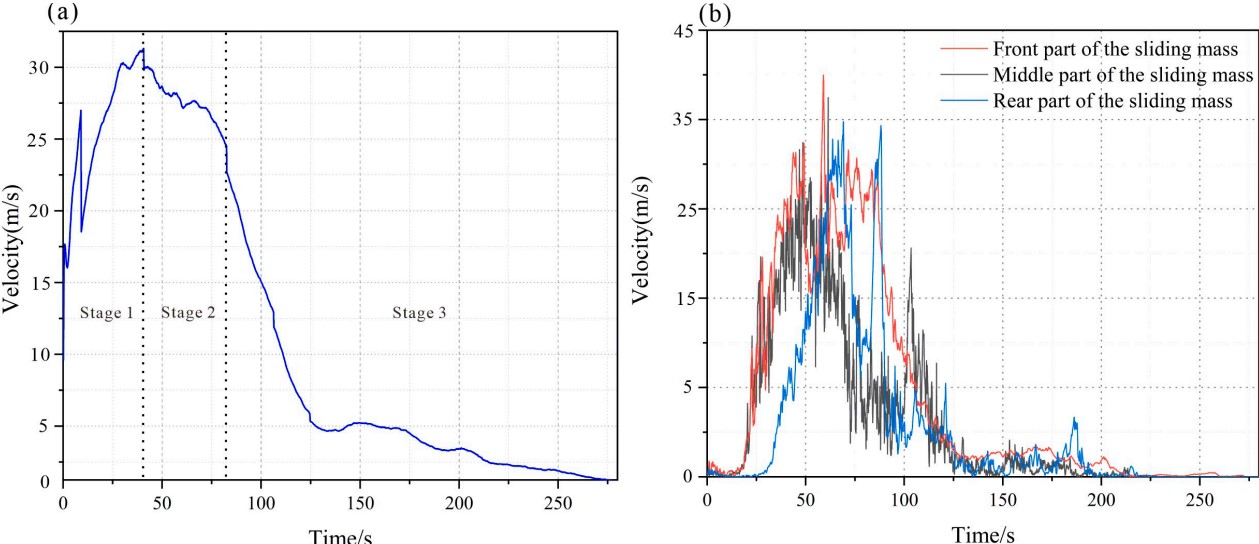

**Figure 7.** The velocity of the rock mass (**a**) The variation curves of the average velocity (**b**) The velocity curves of the velocity in different parts.

After the initiation of the landslide, the rock mass collapsed and broke under the action of gravity, and the velocity increased rapidly before passing through the terrain deflection area. The landslide decayed twice at 10 s and 40 s mainly under the influence of the terrain. After 40 s, the landslide mass passed through the terrain deflection area, and the velocity increased to its peak of 32 m/s. Influenced by the topography of the trench deflection area, the geotechnical body and the mountain body collided fiercely. At T = 125 s, most of the geotechnical body of the landslide reached the trench deflection area and the velocity dropped to 5.1 m/s. As most of the landslide rock mass passed through the trench deflection area, TZRA moved to the relatively wider and flatter terrain, and the speed gradually decreased under the action of friction and collision factors. At approximately 200 s, the landslide stopped accumulating, ending the landslide movement.

Combined with the evolutionary characteristics of the velocity and acceleration distribution of the TZRA movement, avalanche debris can be divided into three main stages, namely, the rapid acceleration stage, the high-speed long-runout, and the final low-speed deposition.

(1) Stage 1: Rapid acceleration Figure 8b: According to the average velocity curve drawn from the data of the monitoring points set at the leading and trailing edges of the landslide (shown in Figure 9), the duration of the rapid acceleration phase of the avalanche debris was 40 s. The average velocity was 32 m/s at the leading edge of the

sliding mass, which was greater than that of 35 m/s at the trailing edge of the sliding mass (Figure 9a). The leading edge of the sliding mass was more influenced by the topography, where two abrupt velocity changes were experienced, mainly due to the deflection area on the three-dimensional topography.

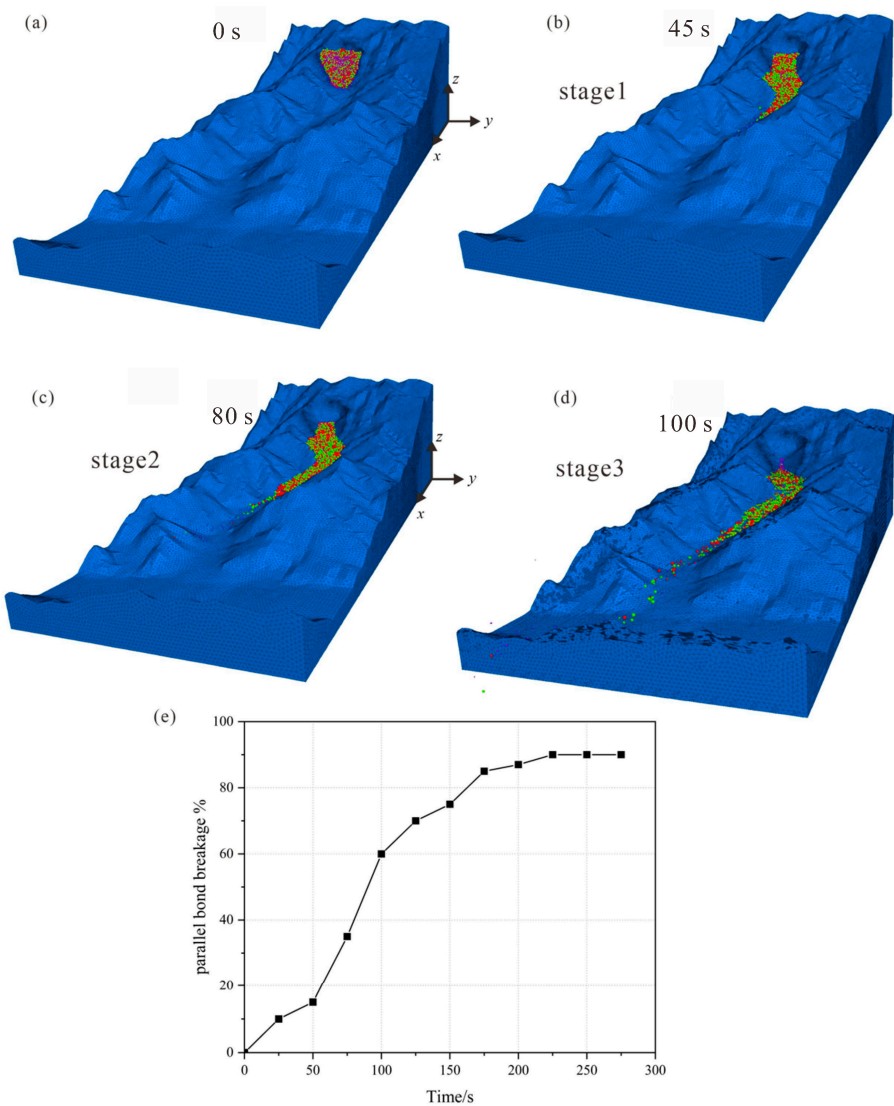

**Figure 8.** (**a–d**) Evolution of the TZRA shown by particle positions at indicated model times; (**e**) Changes of parallel bond breakage with time.

(2) Stage 2: High-speed long-runout Figure 8c: The duration of the high-speed long-runout phase of the avalanche debris was 45–80 s, where the leading edge of the sliding mass lasted for 20s. The avalanche debris is subject to friction and collision, which results in a decrease in velocity. In this stage, the avalanche debris acceleration curve grew slowly due to the continuous collisional disintegration of rock mass and terrain to form avalanche debris that maintained a high speed in the gully.

(3) Stage 3: Final low-speed deposition (Figure 8d): After passing through the last deflection zone, the terrain was flat, and the movement of the particle flow was no longer constrained by the valley. The movement of the particle flow no longer appeared in the middle of the particles due to being wrapped by the friction and collision of the smaller energy consumption and thus maintaining a higher speed. The influence of friction and collision gradually decelerated the particle flow until it stopped at the accumulation. When the TZRA moved to the accumulation area, it

entered the slow deceleration stage, and the avalanche debris moved to an open area altogether, tending the velocity of each part of the geotechnical body toward zero.

(a)

(b)

**Figure 9.** The velocity of the rock mass (**a**) The variation curves of the surface velocity (**b**) The variation curves of the bottom velocity.

In the rapid acceleration stage, the particle movement speed was basically the same, and the speed was at its peak in the deflection area (Figure 9a). Due to the fierce collision between the rock and the mountain, the rock mass disintegrated, and the avalanche debris continued to move forward until it gradually stopped accumulating. The distance of particle movement on the surface was larger than the distance of particle movement at the bottom, which was less than the average distance of particle movement (Figure 9b). This was because the particles on the surface after the initiation of the landslide had the greatest acceleration under the action of gravity and reached the deflection zone first, collided with the mountain, scraped, and shoveled to cause a climbing effect, and thereby dissipated more energy, showing the phenomenon of the shortest movement distance. The rock at the bottom mainly rubbed against the mountain at the bottom, causing a large dissipation of kinetic energy, resulting in a small movement distance. At the same time, from the curve change trend in Figure 8e, it can be inferred that the parallel contact between most of the bonded particles is constantly broken. After the rock avalanche debris began to move for 150 s, the percentage of bond failure between particles reached 78%, and the sliding body was seriously fragmented, so the diffusion and movement range of the whole avalanche debris was great.

*4.2. Simulation of Accumulation Characteristics*

The diffusion mechanism of rock avalanche-avalanche debris in Touzhai was inferred by combining the results of particle size composition of different grain groups, particle morphological characteristics analysis, and field investigation of the stacked body profile. The rock avalanche, in the process of leaving the locked section to form avalanche debris and the final diffusion of its internal compressed air trapped in the graded continuous debris, generates a super-porous gas pressure, weakens the connection between the internal particles, and results in particle suspension (as shown in Figure 10a). The presence of compressed air between the debris particles is the main reason for the particle grinding and dynamic crushing action, and the accumulation does not show the common phenomenon of particles gradually becoming finer from upstream to downstream (as shown in Figure 10b). The high-speed diffusion of the debris compresses the air between the particles, generating super-porous gas pressure. The super-porous gas pressure and the weathering-generated clay minerals intersperse between the debris of each particle size, playing an active role in

trapping compressed air, both of which cause the avalanche debris to remain in motion at very high speeds in a trench with a low apparent friction coefficient.

(a)

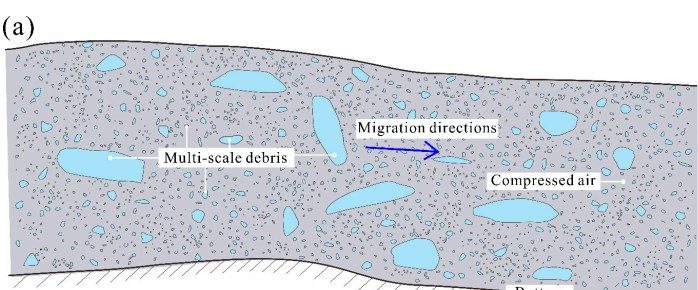

(b)

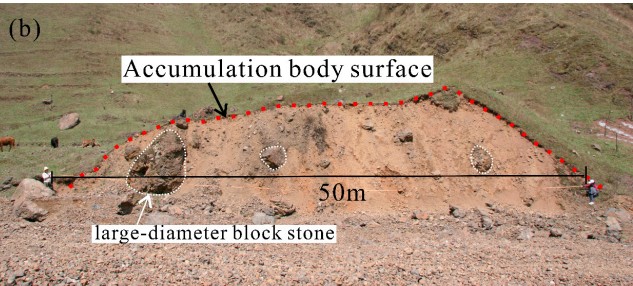

**Figure 10.** (**a**) Sketch of the TZRA during the high-speed propagation of the debris (**b**) Rock block in excavation section of accumulation body.

Overall analysis of the TZRA model along its cross and longitudinal sections deduced that in the process of landslide avalanche debris movement, many particles stop accumulating behind the raised boulders due to the topographic obstruction (Figure 11b), and violent collisions and scraping of the mountains also occur on both sides during the actual TZRA movement. The study established the mountain topography using a wall group as the motion boundary, so the simulation cannot observe scraping and scraping effects on the mountain. However, the accumulation of particles at the location illustrates the validity of the simulation. Figure 11a shows the accumulation of avalanche debris particles, and Figure 11c shows the distribution of the boulders in the entire equivalent particle size range of 2.0–3.0 m in the Touzhai gully. Since the percentage of particles in the particle size range of 1.5–3.0 m in the sphere model established in the PFC$^{3D}$ numerical simulation reached 50%, the simulation results of this particle size range were selected to compare with the actual results. Figure 11b shows that some particles stay on both sides of the slope due to the influence of the topography during the downward transport of particles, while the distribution of scattered boulders on the slope surface can be clearly seen in the survey distribution map of boulders with equivalent particle sizes of 2.0–3.0 m (Figure 11c).

Figures 12 and 13 are the horizontal and vertical sections of the deposit, sectioned according to the positions shown in Figure 4. In the cross-section (shown in Figure 13), the thickness of the accumulation underwent a gradual change from thin to thick from upstream to downstream, and more fine particles remained in the cross-section below the upstream shear outlet (profile). The coarse particles were mainly concentrated in the middle and lower reaches of the gully, which is basically consistent with the results of the boulder distribution survey of the actual site accumulation, and this phenomenon was caused by the larger kinetic energy of large particle-size debris. Simultaneously, it should be noted that the cross-sections of the debris piles with larger thicknesses (as shown in Figure 12) showed that the coarse debris was not only concentrated on the surface of the piles but also existed in the lower part of the piles, and this phenomenon of mixed accumulation of particles was apparent. The accumulation thickness gradually decreased in each cross-section, and finally, the debris rushed out of the trench and spread laterally to form the accumulation fan, which had an overall "trumpet shape" (Debris flow accumulations present an elongated posterior margin and an open anterior margin.) (as shown in Figure 11). The shape of this accumulation fan was slightly different from the rounded edge shape of the field accumulation fan, and the results of the numerical simulation and distribution of the boulder survey were in good agreement. The images of the whole numerical calculation visually showed that the debris accumulation tends to be distributed downstream, and the thickness of the accumulation at the profile gradually becomes thinner, similar to the "planar elongation movement" of the debris in the source area. The longitudinal profiles at different locations showed that the accumulation does not show apparent particle sorting from above and below (Figure 13).

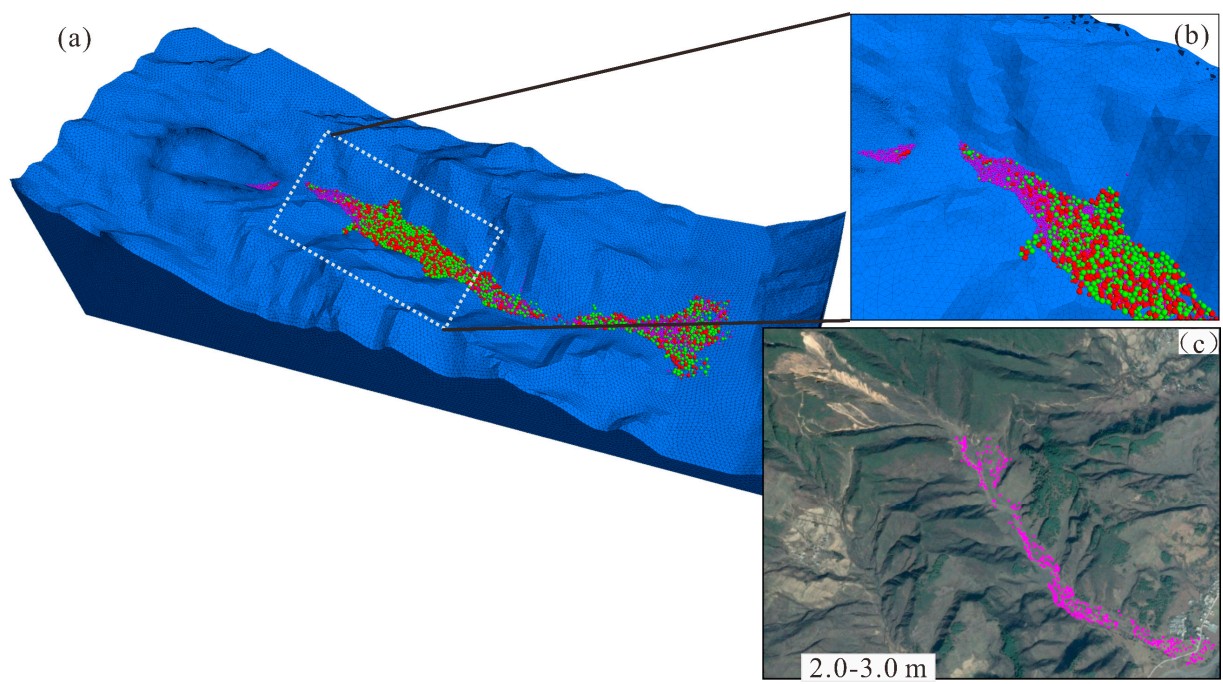

**Figure 11.** (**a**) Numerical simulation of motion accumulation diagram and distribution comparison diagram of boulder equivalent particle size 2.0~3.0 m. (**b**) Local enlargement. (**c**) Map of the distribution of real equivalent grain size 2–3m blocks.

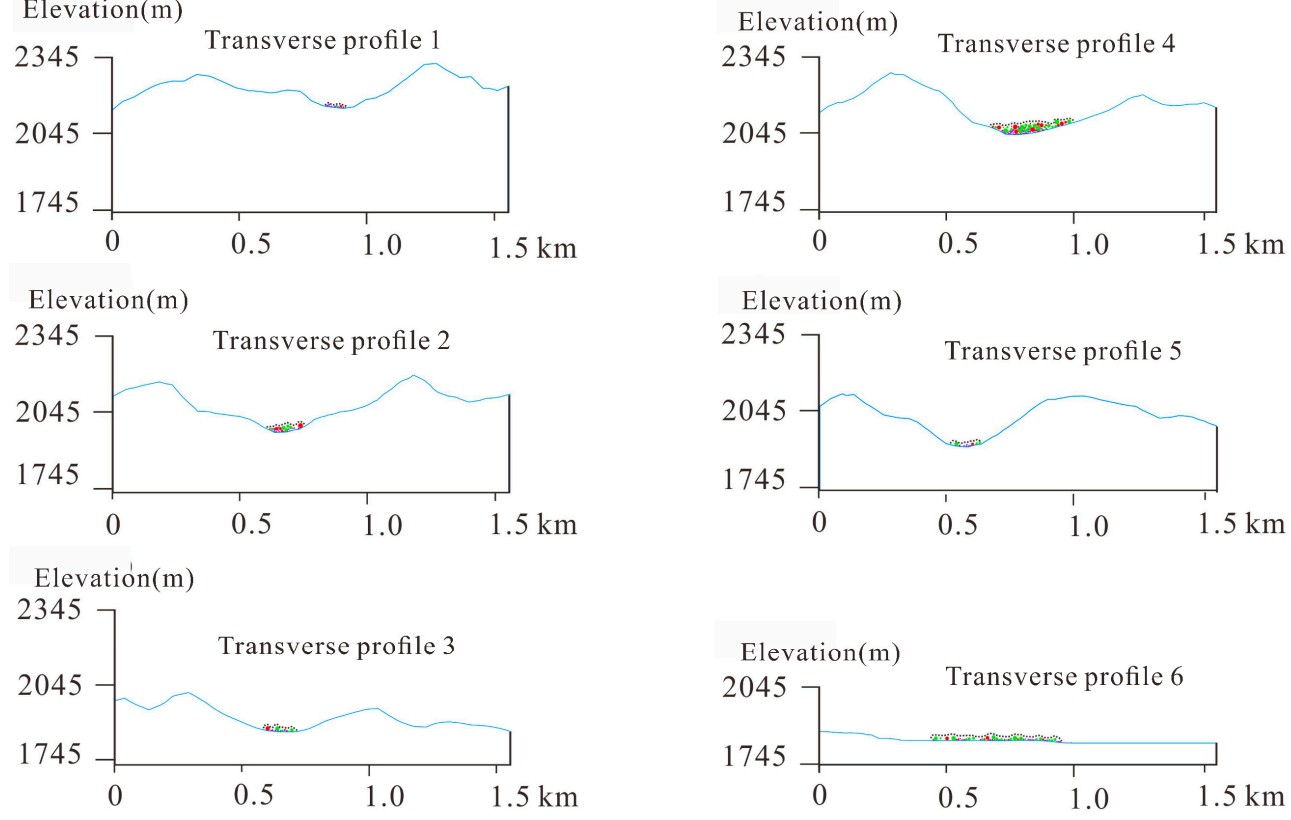

**Figure 12.** Cross section of debris accumulation morphology.

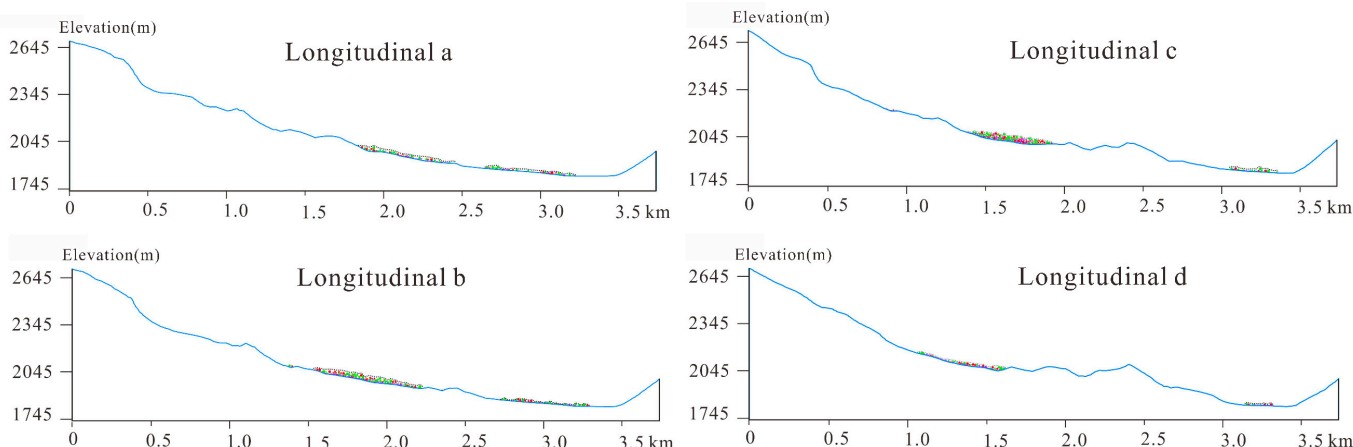

**Figure 13.** Longitudinal section of debris accumulation morphology.

### 4.3. Simulation of Kinematic Process

During the flow of the debris stream, gravitational potential energy is converted into kinetic and deformation energies, which are then dissipated by friction and collision. The energy change of the landslide throughout its movement was analyzed by monitoring the kinetic energy change in the slide (as shown in Figure 14). The moment of energy change inflection point was consistent with those of the mean velocity change curve, confirming that energy decay was the main cause of the substantial decrease in the mean velocity. The avalanche debris contained huge gravitational potential energy at the time of initiation, and the kinetic energy of the landslide was increasing with the velocity. With the gradual movement of the avalanche debris into the valley and the continuous collisional disintegration with the bottom of the slope, the kinetic energy reached its peak after 45 s and gradually began to decay. Simultaneously, as part of the slide stopped moving after colliding with the terrain or the occurrence of climbing, with the gradual increase in friction energy, the avalanche debris moved all the way down along the movement path of the gully and gradually accumulated. The kinetic energy of the landslide body tended to zero out after 180 s, when the avalanche debris reached the mouth of the trench in the accumulation and diffusion stage. The energy monitoring curve was roughly similar in time to the inflection point of the average velocity curve, where the decrease in velocity and the decay of energy were synchronized, from a linear increase at the beginning to a gradual decay and finally convergence to zero. The avalanche debris was subjected to intense friction as it moved along the head of the gully due to the influence of the valley topography. The accumulated frictional energy of the slide body formed an upward trend, and the inflection point of the curve appeared at 90 s (as shown in Figure 14b). The avalanche debris had a large number of particles that stopped moving at 120 s, and eventually, all the gravitational potential energy was dissipated by collision and friction.

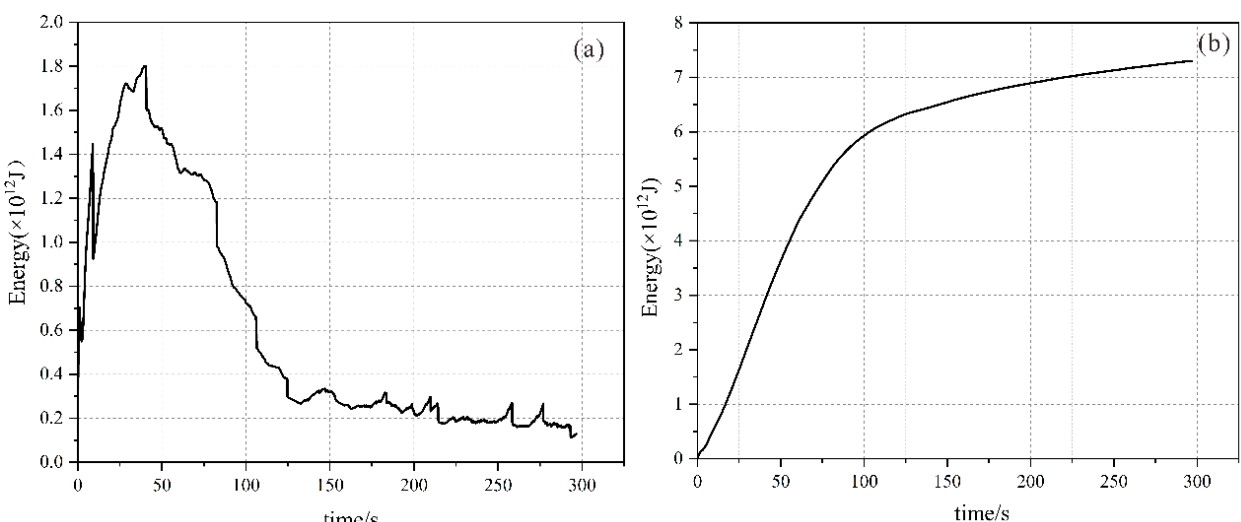

**Figure 14.** (**a**) Sliding body kinetic energy curve. (**b**) Cumulative friction energy dissipation curve.

## 5. Discussion

### 5.1. Mechanisms for the Occurrence of Rock Avalanche

Fluidization is the primary mode of movement for high-speed, long-runout landslides following debris disintegration, and it is also a prerequisite for further amplification of their disastrous effects. Heim [1] was the first to systematically describe the fluidization of high-speed, long-distance landslides during his study of the Elm landslide in 1881. After the disintegration, landslides move at high speed in the form of avalanche debris, exhibiting fluid-like behavior such as climbing and avoiding obstacles in their path. When studying high-speed, long-runout landslides using the discrete element method, it is crucial to reproduce the motion process and accumulation characteristics. To achieve this, researchers have employed various simulation techniques. Tang et al. [7] used a particle parallel bond model to analyze the kinematic characteristics of the Jiufenershan landslide induced by an earthquake. Their simulation results were found to be in line with the accumulation characteristics of real landslides, particularly under low residual friction coefficient and medium particle bond strength conditions. Furthermore, the exploration of the long run-out mechanism of a landslide demonstrated that self-lubrication, frictional vaporization, and frictional melting effects might be the main reasons for the reduced frictional resistance of a landslide [45–47]. To study typical high-speed, long run-out landslides such as the Donghekou and Wenjiagou landslides triggered by the Wenchuan earthquake, a discrete element method based on the parallel cohesive model can better simulate the debrisisation and fluidization motion process of landslides, reproducing their typical accumulation characteristics, and revealing the destabilization and motion mechanism [48–52].

### 5.2. Effect of the Topography

The geomorphic landscape after the TZRA occurrence is different from the current geomorphic landscape. The sorting of the accumulation is poor, and there is no apparent particle sorting phenomenon. Simultaneously, large-size stones still exist in the middle of the deposited profile rather than being concentrated on the surface of the deposit. Therefore, the 'reverse grading' phenomenon is not apparent in the TZRA deposit. Combined results of the particle size composition of different grain groups, particle morphology characteristics analysis, and field investigation of the mound profile inferred the diffusion mechanism of the TZRA. In the process of diffusion, the compressed air in the accumulation body is trapped in the graded continuous debris, resulting in excess pore gas pressure, weakening the connection between the particles, and resulting in particle suspension. This is precisely because the compressed air trapped between the debris particles exerts a cohesive force between the particles to weaken them, resulting in weak particle grinding and dynamic

crushing, and the debris of the deposit can still maintain a high roundness, so the phenomenon of particles gradually becoming finer from upstream to downstream does not occur. In discrete element numerical simulations, the particles in the middle were found to be greater than the average value in terms of movement distance and velocity. Since this part of the particles is less influenced by the topography, it causes the long-runout transport of the avalanche debris. The terrain has a great influence on the accumulation of morphological characteristics and the transport distance of the avalanche debris, which should not be neglected because of the large decay in velocity after passing through both deflection zones.

*5.3. Integration of Multi-Source Analysis Techniques*

The discrete element model (DEM) is an approach that considers the sliding body as a collection of particle flows. This method provides a more accurate simulation of various aspects, including the impact of the sliding body, scraping effect, mutual collision, and rolling between particles. It also allows for the reverse analysis of the movement process of the landslide debris flow.

However, the topography has a great influence on the movement speed and diffusion range of rock avalanche debris flow. Additionally, the application of GIS and remote sensing technology plays a crucial role in identifying and delineating areas affected by the movement and accumulation of rock avalanche debris flow. This inversion model's accuracy is validated through these advancements. Precise mapping of topographic features using high-resolution remote sensing enables the detailed modeling of individual elements, which is essential for understanding the process behind rock avalanche debris flow.

In recent years, significant progress has been made in identifying slippery geological structures, delineating landslide boundaries, and classifying the evolution stages of landslides, thanks to advancements in Unmanned Air Vehicles (UAVs), optical remote sensing image satellites, and interferometric synthetic aperture radar [53,54]. However, due to various factors such as topography, climate, rainfall, and vegetation coverage specific to the study area, the understanding of the movement process for high-speed long-runout rock avalanche debris flows is still lacking [55,56]. While these technologies can provide early warning and help define the scope of the disaster before and after its occurrence, they are unable to capture the detailed dynamics of the movement.

Future research can be carried out from the 'space-ground' integrated remote sensing monitoring, combined with the landslide movement and diffusion accumulation range simulation of DEM. It plays a good role in guiding the subsequent realization of the combination of "geological analysis, multi-source remote sensing analysis and DEM inversion method". It guides the analysis and prediction of the whole process of rockfall-debris flow, from the disaster-generating process to the evolution of the movement and spreading accumulation range.

## 6. Conclusions

The TZRA was a high-speed long-runout landslide with a volume of approximately $9 \times 10^6$ m$^3$ that traveled a vertical height of 1120 m (H) over a distance of 4000 m (L), yielding an effective friction coefficient of approximately 0.22 (H/L). A combination of field investigation and discrete element simulation was used to investigate the motion and dispersion characteristics of the TZRA in Yunnan Province, China. The important findings of the study can be summarized as follows:

(1) The kinetic phase of the TZRA probably lasted for 3 min and can be further divided into three phases: the rapid acceleration (<45 s) phase; the high-speed long-runout (45–120 s) phase and the final low-speed deposition (>2 min) phase.

(2) Analysis of the accumulation pattern of the TZRA indicated that during the movement of the landslide-avalanche debris, a climbing phenomenon and scraping effect coexisted on both sides of the mountain. When the TZRA reached the mouth of the gully, the avalanche debris spread to both sides because it was no longer restrained by

the mountain bodies on both sides of the narrow gully, forming a "trumpet"-shaped accumulation pattern that is roughly consistent with the results of the large-size boulder survey.

(3) Numerical results show that the debris accumulation tends to be distributed downstream, and the accumulation thickness gradually thins at the profile, similar to the "planar elongation movement" of debris in the source area; therefore, accumulation does not show evident particle sorting from above and below.

**Author Contributions:** Conceptualization, X.L.; methodology, Y.G.; software, Y.G., Y.L. (Yanan Li) and X.X.; validation, C.Z. and X.X.; formal analysis, H.L. and Y.L. (Yanlin Li); investigation, Y.G. and Y.L. (Yanan Li); resources, X.X. and C.Z.; data curation, Y.G. and X.X.; writing—original draft, Y.G. and X.X.; writing—review and editing, Y.G. and C.Z.; visualization, J.Y.; supervision, X.L.; project administration, X.L.; funding acquisition, X.L. and C.Z. All authors have read and agreed to the published version of the manuscript.

**Funding:** The research work was funded by the Research Fund of National Natural Science Foundation of China (NSFC) (Grant No. 42277154), National Natural Science Foundation of Shandong Province of China (NSFC) (Grant No. ZR2022ME188), Jinan City "new university 20" research leader studio project (Grant No. 20228108), the opening fund of Key Laboratory of Geohazard Prevention of Hilly Mountains, Ministry of Natural Resources (Grant No. FJKLGH2023K001). Thanks for providing some of the data provided by Zemin Xu of Kunming University of Science and Technology.

**Data Availability Statement:** Most of the data generated during this study are included in the article. For other datasets, please contact the corresponding author with reasonable requests.

**Conflicts of Interest:** The authors declare no conflict of interest.

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
