# Peer review of "Insights into the Movement and Diffusion Accumulation Characteristics of a Catastrophic Rock Avalanche Debris—A Case Study"

_remotesensing, doi:10.3390/rs15215154_

Round 1

Reviewer 1 Report

The manuscript investigates the movement and diffusion accumulation characteristics of a rock avalanche based on field investigation and discrete element simulation. Reliable discrete element landslide hazard inversion methods proposed and verified by measured data. The paper is very concise and well written. The figures are clear, although the figure captions need considerable expansion to provide more necessary information.

The manuscript requires minor revision.

Ⅰ: The paper begins with a thorough introduction, which gives a good overview over the scientific field and motivates the development of the presented method, including a brief comparison to other existing methods. Some cases should be improved.

1) The title of the manuscript is too long and the main idea of the manuscript is not clear enough, so it is recommended to revise it.

2) Besides empirical and numerical methods, physical modeling experiment is also an effective approach for investigating the formation process of dispersal and accumulation of rockfalls and debris flows. Hope the authors can add a brief summary of the advances in experiment methods.

3) This paper provides a review of the classic literature in this field. Some recent published papers are missing in the literature review of numerical methods on debris-flow inversion events.

Ⅱ: The following issues need address in the part of Methodology.

1) Section 3.1 lines 241,244,247 P7: In the DEM model the author mentions 'wall' and the author does not make a clear note of what he means.

2) Section 3.2 P8: Whether the selection of DEM model parameters needs to be calculated is not stated in the article.

3) Section 3.2 lines 286 P9: For the principles of parameter selection in the DEM model, the paper does not provide a detailed explanation, such as the shear stiffness and normal stiffness in Table 3.

4) Section 3.2: lines 292, P9: The author should explain "merical-experimental data" in detail. It might be better to show data or similar for a detailed explanation of Numerical simulations.

: The Simulation results part appears to thorough, using both field observations and numerical methods. The conclusions are convincing. However, several points should be further explained.

1) Section 4.2 Fig.11 P13: How to determine the results using GPS so that readers can understand. The author should describe the accuracy of this result.

2) Section 4.2 lines 426: For the shape of the buildup of a rockslide debris flow the authors use the term about the "trumpet shape". In order to be easily understood by the reader, the author needs to provide a detailed explanation. 

NUll

Reviewer 2 Report

The paper studied the dynamic process and diffusion mechanism of the 1991 Touzhai long runout rock avalanche using field investigation and 3D discrete element modelling. Several new findings are obtained from numerical simulation, which bring some input to the scientific debate. The topic is interesting, and the paper is well organized. In general, the paper contains all the required elements, but it is a bit superficial in the introduction part. Furthermore, the authors need to check carefully through the manuscript to fix all typos and mistakes. I suggest minor revisions in order to improve the quality of the paper.The comments for this study are listed below:

*Abstract.

1. Lines 21~23: Sentences are gramatically incomplete or wrong.

*Introduction.

2. Lines 99~119: Continuum-based methods such as smoothed particle hydrodynamics (SPH) and discontinuum-based methods such as discontinuous deformation analysis (DDA) are also widely used for landslide numerical simulation. However, only the discrete element method (DEM) is introduced in the literature review of numerical techniques. It is not the standard and a more thorough literature review should be added. In addition, the advantages of the DEM should also be identified.

3. Lines 120~127: The limitations of previous studies and the motivation of the current study should be summarized.

* Study area.

4. Line 130~131: Panhe River, Panhe Township, Zhaotong, Yunnan ===>>> Panhe River, Yunnan province, China

5. Figure 2: original forms ===>>> Pre-failure landform; landslide deposit ===>>> Landslide deposit

6. Figure 3: 180cm ===>>> 180 cm

* Methodology

7. Line 225: Figures 4 ===>>> Figure 4

8. 3.1 Discrete element modelling: A brief introduction on the DEM is suggested to added.

9. Line 225: Figures 5: the longitudinal profiles a~b and the transverse profiles 1~6 should be explained in the figure caption.

10. Table 3: Gpa ===>>> GPa; Mpa ===>>> MPa

* Simulation results

11. Figure 8: 0s ===>>> 0 s; 45S ===>>> 45 s

12. Fig. 12: plane 1 ===>>> Transverse profile 1

13. Fig. 13: plane a ===>>> Longitudinal profile a

14. Fig. 14a: 1012 ===>>> 1012 J

 Minor editing of English language required

Reviewer 3 Report

This paper presents the results of a retrospective analysis concerning the mechanism of occurrence and the flow of a landslide that happened in Touzhai in 1991. While it provides valuable data on landslides and conducts an analysis focusing on grain size using DEM, the content is intriguing in terms of its utility and novelty. However, for the sake of rigor, there remain certain issues that warrant clarification within the manuscript.

Comments 1:P5, L174: Please provide a more detailed explanation about 0.22.

Comments 2:

P6, Fig.3: The red color is hard to see; please adjust. The grain size distribution does not seem to be the typical cumulative grain size curve. Please add an explanation.

Comments 3:

P8, L276: Please provide a detailed explanation about how you determined the UCSm.

Comments 4:

P9, L287-294: It would be better to explain the DEM model in a figure or similar. For example, what exactly are kn and ks? What are their units? What do 5e6 and 2e6 mean?

Comments 5:

The numerical damping parameters are important micro-parameters in the PFC model. Which type of damping is considered in this study?

Comments 6:

P12, L373: The content about "Simulation of accumulation characteristics" was difficult to understand, and the reviewer couldn't grasp it. Especially in comparison between analysis and actual data, it should be clearly stated what matches and what doesn't.

Comments 7:

How did you set the friction coefficient between the slip surface and the sliding soil mass? Didn't this friction significantly affect the flow behavior?

Comments 8:

Fig. 10 is of good quality. How did the authors plot the simulation results in the actual terrain?

The language is good.
